# A Comparison of Hip Muscle Mass, Muscle Power, and Clinical Outcomes with Long-Term Follow-Up in Patients with Metal-on-Metal Hip Arthroplasty Compared to Metal-on-Polyethylene Hip Arthroplasty

Mette Holm Hjorth [1,2], Inger Mechlenburg [1,2,3,*], Frederik Nicolai Foldager [1] , Marianne Tjur [1,2] and Maiken Stilling [1,2]

1   Department of Orthopaedics, Aarhus University Hospital, Palle Juul-Jensens Blv. 99, 8200 Aarhus, Denmark
2   Department of Clinical Medicine, Aarhus University, Palle Juul-Jensens Boulevard 82, 8200 Aarhus, Denmark
3   Department of Public Health, Aarhus University, Bartholins Allé 2, 8000 Aarhus, Denmark
*   Correspondence: inger.mechlenburg@clin.au.dk

**Abstract:** (1) Background: Metal-on-metal (MoM) total hip arthroplasty (THA) and hip resurfacing arthroplasty (HRA) was presumed to provide superior functional outcomes compared to metal-on-polyethylene (MoP) THA. (2) Methods: We compared muscle mass, power, step test asymmetry, and patient-reported outcomes between MoM THA/HRA and MoP THA. A total of 51 MoM THA/HRAs and 23 MoP THAs participated in the cross-sectional study at a mean of 6.5 (2.4–12.5) years postoperatively. Muscle mass was measured by Dual energy X-ray Absorption (DXA) scans and muscle power in a Leg Extensor Power Rig. Step test asymmetry was obtained with an Inertial Measurement Unit (IMU). The patients completed the Harris Hip Score (HHS) and the Copenhagen Hip and Groin Outcome Score (HAGOS). (3) Results: The MoM THA/HRA group had a greater inter-limb difference in hip muscle mass compared to the MoP THA group ($p = 0.02$). Other inter-limb differences in muscle mass and power were similar ($p > 0.05$). Muscle mass of the thigh and calf area and muscle power in both legs were higher in MoM THA/HRA compared to MoP THA ($p < 0.009$). Step test time asymmetry when ascending was lower in MoM THA/HRA compared to MoP THA ($p = 0.03$). HHS and HAGOS scores were similar between groups ($p > 0.05$). (4) Conclusion: Overall, we could not verify the hypothesis that MoM THA/HRA contributes to superior functional outcomes compared to MoP THA.

**Keywords:** metal-on-metal hip arthroplasty; metal-on-polyethylene hip arthroplasty; DXA scan; muscle mass; muscle power; functional outcome test

## 1. Introduction

For decades, end-stage osteoarthritis of the hip has been successfully treated with metal-on-polyethylene (MoP) total hip arthroplasty (THA) [1–3]. A major limitation, especially in young patients, is the polyethylene wear particles produced by the bearing surfaces and associated with osteolysis and aseptic loosening of the implant [4–7]. Metal-on-metal (MoM) THA and hip resurfacing arthroplasty (HRA) were expected to lower dislocation rates and increase range-of-motion and functional capacity as the femoral component mimic the natural human anatomy [8]. MoM THA was previously recommended in younger patients. Furthermore, the MoM THA/HRA was hoped to enable patients to return to pre-osteoarthritic potential. However, due to concerns regarding adverse reaction to metal debris (ARMD), the success for MoM hip articulations was short-term [9–14]. Thus, the use of MoM THA/HRA has fallen to <1% [10,15]. Studies have investigated the short-term differences in patient-reported outcomes, gait patterns, and performance-based tests between patients undergoing MoM THA/HRA and patients undergoing MoP THA [16–20]. A few

of these studies suggest that a better anatomical preservation and a larger femoral head may lead to superior outcomes in performance-based tests [20,21]. However, it is unclear if the potential benefits of MoM THA/HRA with large femoral heads contribute to superior outcome in performance-based tests compared to MoP THA with smaller femoral heads. Furthermore, mid- and long-term results are missing. We reported that even young, active patients undergoing MoM THA had not fully regained muscle mass or muscle power at 5–7 years postoperatively [22]. Additionally, it is unclear whether patients with MoM hip articulations achieve a better recovery of muscle mass and leg power compared to patients with MoP THA at mid- and long-term follow-up. Overall, we wanted to investigate if MoM THA/HRA contributes to superior functional outcomes compared to MoP THA at mid- to long-term follow-up. We investigated three hypotheses: (1) muscle mass and muscle power are higher in patients undergoing MoM THA/HRA compared to patients undergoing MoP THA, (2) the inter-limb difference in muscle mass and muscle power is smaller in patients undergoing MoM THA/HRA compared to patients undergoing MoP THA, and (3) patients undergoing MoM THA/HRA experience less step test asymmetry compared to patients undergoing MoP THA.

## 2. Materials and Methods

### 2.1. Patients and Articulations

In 2014, 111 patients (50 females, 61 males) with a total of 148 THAs participated in a cross-sectional study at Aarhus University Hospital, Denmark, at a mean of 6.5 (2.4–12.5) years postoperatively. The study was conducted in accordance with the Declaration of Helsinki [23] and approved by the Central Denmark Region Committee on Health Research Ethics (jr. nr.: 1-10-72-65-14) and the Danish Data Protection Agency (jr. nr.: 2007-58-0010, Trial nr.: 1-16-02-87-14). Patients were recruited among participants from the department's former research projects on THA. Inclusion criteria were patients undergoing MoM or MoP primary hip arthroplasty and who gave informed consent to participate. A total of 37 patients had bilateral hip articulations which preclude reasonable comparison of inter-limb differences in this study and were thus excluded, resulting in 74 patients with a mean age of 59 (30–77) years. All patients were able to walk without walking aids. Patients were divided into two groups. The finale sample included the MoM THA/HRA group ($n = 51$), including 33 males and 18 females, and the MoP THA group ($n = 23$), including 8 males and 15 females. All MoM THAs ($n = 18$) underwent a posterior surgical approach. The MoM HRAs underwent either a posterior surgical approach ad modum Moore ($n = 22$), or an antero-lateral surgical approach ad modum Watson ($n = 11$). All MoP THAs ($n = 23$) underwent a posterior surgical approach. MoM HRA patients who operated with the posterior approach had a partial detachment of the tendinous insertion of gluteus maximus on the femoral bone. A flowchart of included patients and their articulation types and sizes are presented in Figure 1. The MoM/HRA THA group had a mean femoral head size of 55.2 (50–64) and the MoP THA group had a mean femoral head size of 54.6 (50–62). Baseline characteristics of all patients are presented in Table 1.

### 2.2. Dual-Energy X-ray Absorptiometry (DXA) Scans

DXA scans were performed with a Lunar iDXA 2013 DXA scanner (General Electric Medical Systems, Maddison, WI, USA) and analyses were performed using the enCORE version 16.00 software. Custom regions of interest (ROI) were made as previously described [22] from anatomical fix points on the hip, thigh, and calf, and muscle mass (g) was divided by the area of the ROI ($cm^2$) (Figure 2). The coefficient of variation for the thigh and calf segments have been reported to be between 0.7–1.8% [24,25].

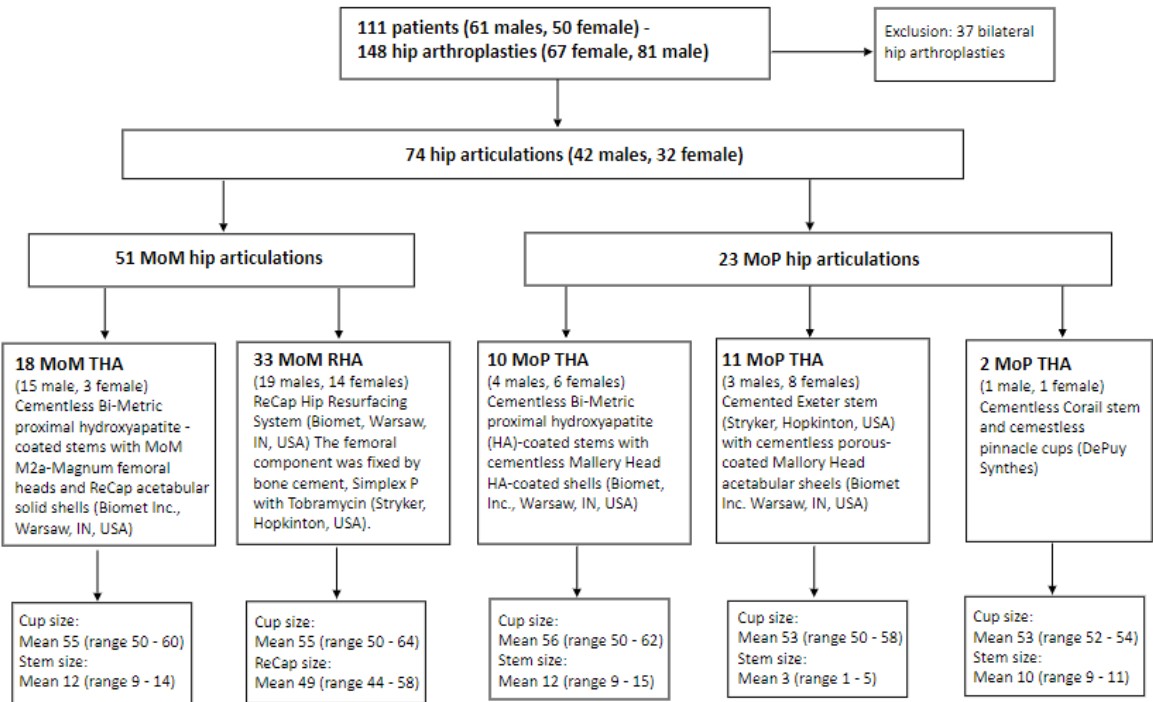

**Figure 1.** Flowchart of included patients and articulation types and sizes.

**Table 1.** Baseline characteristics of patients with metal-on-metal (MoM) total hip articulation (THA)/hip resurfacing articulation (HRA) and metal-on-polyethylene (MoP) THA.

| Articulation | MoM THA/HRA | MoP THA | *p*-Value [a] |
|---|---|---|---|
| Number of patients | 51 | 23 | - |
| Sex (male/female) | 33/18 | 8/15 | 0.02 |
| Age at follow-up, mean (range) | 56 (30–71) | 67 (45–77) | 0.00 |
| Years since operation, mean (range) | 5.8 (2.4–9.0) | 8.5 (6–12.5) | 0.00 |
| BMI (kg/m$^2$) (range) | 23.7 (17–34) | 22.6 (16–30) | 0.20 |
| Implant side, right/left | 31/20 | 11/12 | 0.30 |

[a] Satterthwaite's *t*-test. BMI = body mass index.

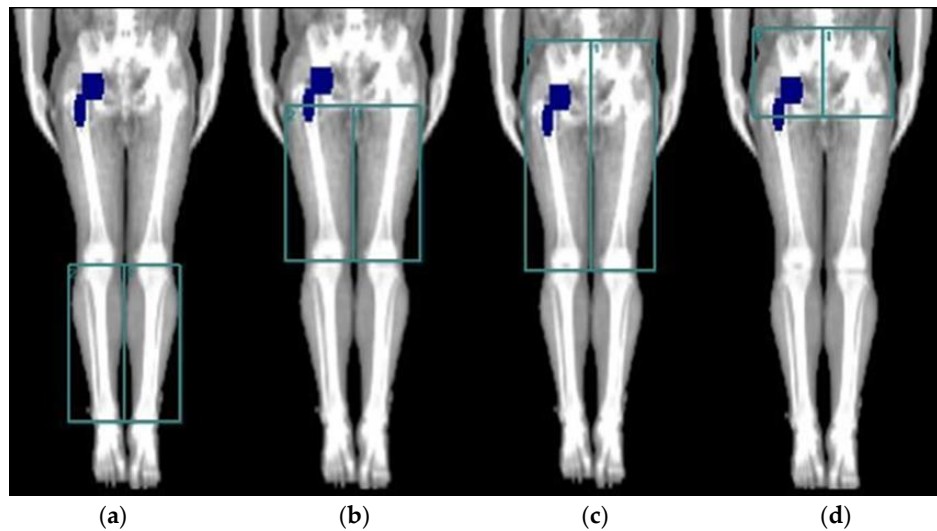

(a)      (b)      (c)      (d)

**Figure 2.** The four different DXA scan regions of interest (ROIs) used for evaluation of mean muscle mass of the implant-side and the non-implant-side leg. (**a**) calf area, (**b**) thigh area, (**c**) hip and thigh area, and (**d**) hip area.

### 2.3. Leg Extensor Power Rig (LEPR)

Patients performed a mean of 6.4 (3–10) explosive leg extensions with the implant side leg and a mean of 5.8 (3–10) explosive leg extensions with the non-implant side leg until no further improvements were seen. The best recorded power output was used for statistical analysis. One patient in the MoM HRA group had recently had knee surgery and was unable to complete the explosive leg extensions in the LEPR. LEPR measurements were performed seated using one leg at a time where patients pushed a footplate as hard and fast as possible [22]. Measures of LEPR were summarized as relative power using Equation (1):

$$\text{Relative power} \left(\frac{\text{W}}{\text{Kg}}\right) = \frac{\text{Absolute power (W)}}{\text{Body Weigth (Kg)}} \tag{1}$$

The LEPR used in patients with end-stage osteoarthritis of the hip and in THA patients has a fair to good reliability with intraclass correlation coefficients above 0.72 [26,27]. Because the muscles in the hip and thigh areas mainly generate the power estimated in the LEPR, correlations between mean muscle mass and mean muscle power were performed using the mean muscle mass of the hip and thigh areas.

### 2.4. Inertial Measurement Unit (IMU)

The IMU (MicroStrain, Inertia-Link-3DM-GX2, Williston, ND, USA) measures orientation, velocity, and gravitational forces in a three-dimensional space, using a combination of a gyroscope and an accelerometer [28,29]. The IMU was carefully fixed to the skin over the sacrum with double adhesive tape. Data were collected via a wireless Bluetooth connection, and real-time data from the sensor were stored on a computer with a sampling frequency of 100 Hz. Data analysis of the step test parameters was performed by proprietary, non-disclosed algorithms provided by the manufacturer based on the algorithms by Zijlstra et al. [30]. Inertial sensors have shown potential in functional tests [31,32], and stepping has been shown to be a powerful performance-based test with good discriminate capacity to differentiate between healthy subjects and patients with osteoarthritis [31]. The IMU-based method has been shown to be reliable and reproducible in assessing gait and performance-based tests in healthy subjects [33,34], and has also shown potential for performance-based tests such as ascending–descending movements in step tests and sit-to-stand transfers [22,31,35,36]. Asymmetry scores (ASs) between the implant-side leg (ASi) and the non-implant-side leg (ASn) were calculated for each parameter using Equation (2):

$$\text{ASs} = 100\% \times \left(\left(\frac{\text{ASi} - \text{ASn}}{\text{ASi} + \text{ASn}}\right)/2\right) \tag{2}$$

### 2.5. Step Test

All patients were tested on a 40 cm high step bench, and if too challenging, they were offered a 30 cm high step bench. Patients were asked to ascend and descend the step bench at their own pace three times with each leg, always beginning the first step-up with the non-implant-side leg. The IMU parameters representing the average of the three repetitions are based on peak detection between the start and the end of each test.

### 2.6. Harris Hip Score (HHS) and the Copenhagen Hip and Groin Outcome Score (HAGOS)

All patients were examined according to the surgeon-reported HHS (range 0–100) [37], and completed the patient-reported HAGOS questionnaire (range 0–100) [38]. The HHS covers domains of pain, daily activities (stair climbing, public transportation, sitting, and putting on socks and shoes), gait (limp, support needed, and walking distance), and range of motion. Intraclass correlation coefficients have been proven good to excellent ranging from 0.74–1.00 [39]. The HAGOS consists of six separate subscales assessing Pain, Symptoms, Physical function in daily living, Physical function in Sport and Recreation, Participation in Physical Activities, and hip and/or groin-related Quality of Life (QOL).

The questionnaire has shown intraclass correlation coefficients ranging from 0.82 to 0.91 for the six subscales [38].

### 2.7. Statistical Analysis

All continuous variables were tested for normality using the Shapiro–Wilk test [40]. When data were not normally distributed, we used the non-parametric tests Mann–Whitney U-test and Wilcoxon rank-sum test. When data were normally distributed, we used a paired *t*-test. Calculations of the correlation coefficient (r) of independent variables were made using Spearman correlation analysis when data were not normally distributed, and Pearson's correlations analysis was used when data were normally distributed. *p*-values below 0.05 were considered statistically significant. All analyses were performed using STATA software version 13 (StataCorp LP, College Station, TX, USA).

## 3. Results

### 3.1. Dual-Energy X-ray Absorptiometry (DXA) Scans

Patients undergoing MoM THA/HRA had significantly more muscle mass at both the implant side and the non-implant side leg compared to patients undergoing MoP THA ($p < 0.05$). This trend was found in all areas of the examined legs, except in the hip area at the implant side leg, where measurements of muscle mass were similar ($p = 0.24$). Measurements of the inter-limb difference in muscle mass were different between groups in the hip area ($p = 0.02$). The inter-limb differences in muscle mass of the thigh, calf, and the hip and thigh area were similar ($p > 0.05$) (Table 2).

**Table 2.** Comparison of mean muscle mass in both legs and inter-limb difference between mean muscle mass in MoM THA/HRA ($n = 51$) and MoP THA ($n = 23$). Values are mean (SD).

| Region of Interest (ROI) | MoM THA/HRA | MoP THA | *p*-Value |
|---|---|---|---|
| Hip area (g/cm$^2$) | | | |
|     Implant side | 9.44 (1.50) | 8.98 (1.59) | 0.24 [b] |
|     Non-implant side | 10.63 (1.57) | 9.78 (1.55) | 0.04 [b] |
|     Inter-limb difference | 1.28 (0.71) | 0.92 (0.47) | 0.02 [a] |
| Thigh area (g/cm$^2$) | | | |
|     Implant side | 6.96 (1.19) | 5.83 (1.28) | 0.0004 [b] |
|     Non-implant side | 7.13 (1.26) | 6.04 (1.22) | 0.0008 [b] |
|     Inter-limb difference | 0.38 (0.33) | 0.41 (0.40) | 0.71 [a] |
| Hip and thigh area (g/cm$^2$) | | | |
|     Implant side | 7.88 (1.15) | 6.71 (1.39) | 0.0003 [b] |
|     Non-implant side | s8.35 (1.18) | 7.20 (1.58) | 0.0008 [b] |
|     Inter-limb difference | 0.51 (0.34) | 0.54 (0.37) | 0.76 [a] |
| Calf area (g/cm$^2$) | | | |
|     Implant side | 3.63 (0.53) | 3.35 (1.01) | 0.0038 [b] |
|     Non-implant side | 3.69 (0.49) | 3.47 (1.07) | 0.0097 [b] |
|     Inter-limb difference | 0.29 (0.21) | 0.18 (0.12) | 0.06 [a] |

[a] Two-sample Wilcoxon rank-sum (Mann–Whitney) test. [b] Paired *t*-test.

### 3.2. Leg Extensor Power Rig (LEPR)

Patients undergoing MoM THA/HRA had significantly more power in both the implant side leg ($p < 0.000$) and the non-implant side leg ($p < 0.002$) compared to patients undergoing MoP THA, but the inter-limb difference in muscle power between groups was not significant ($p = 0.87$) (Table 3). Mean muscle power in the two groups correlated in the following way with the mean hip and thigh muscle mass. The implant-side leg in the MoM THA/HRA group ($r = 0.54$, $p = 0.000$). The implant-side leg in in the MoP THA group ($r = 0.06$, $p = 0.39$). The non-implant-side leg in the MoM THA/HRA group ($r = 0.54$, $p = 0.000$). The non-implant-side leg in the MoP THA group ($r = 0.30$, $p = 0.16$) (Figure 3a,b).

**Table 3.** Comparison of mean muscle power in both legs, inter-limb difference in mean muscle power, Harris Hip score (HHS), and the Copenhagen Hip and Groin Outcome Score (HAGOS) between MoM THA/RHA (*n* = 50) and MoP THA (*n* = 23). Values are mean (SD).

|  | MoM THA/RHA | MoP THA | *p*-Value [a] |
|---|---|---|---|
| **Power (W/kg)** | | | |
| Implant side | 1.98 (0.67) | 1.29 (0.50) | 0.000 |
| Non-implant side | 2.02 (0.69) | 1.50 (0.50) | 0.002 |
| Inter-limb difference | 0.24 (0.18) | 0.23 (0.18) | 0.87 |
| **Power (W)** | | | |
| Implant side | 164.06 (61.55) | 100.96 (45.19) | 0.000 |
| Non-implant side | 168.08 (62.41) | 115.78 (43.71) | 0.0003 |
| Inter-limb difference | 19.18 (14.47) | 17.35 (11.79) | 0.75 |
| **HHS** | | | |
| Score | 97.7 (4.2) | 96.6 (5.2) | 0.26 |
| **HAGOS** | | | |
| Symptoms | 90.5 (13.0) | 84.8 (16.9) | 0.18 |
| Pain | 93.9 (10.2) | 85.3 (21.9) | 0.07 |
| Function in Daily Living | 91.7 (15.6) | 85.9 (18.7) | 0.08 |
| Sport and recreation | 83.8 (20.5) | 72.9 (27.1) | 0.08 |
| Physical Activities | 78.9 (29.4) | 79.4 (26.6) | 0.99 |
| Hip related Quality of Life | 81.8 (21.3) | 77.4 (26.5) | 0.68 |

[a] Two-sample Wilcoxon rank-sum (Mann–Whitney) test.

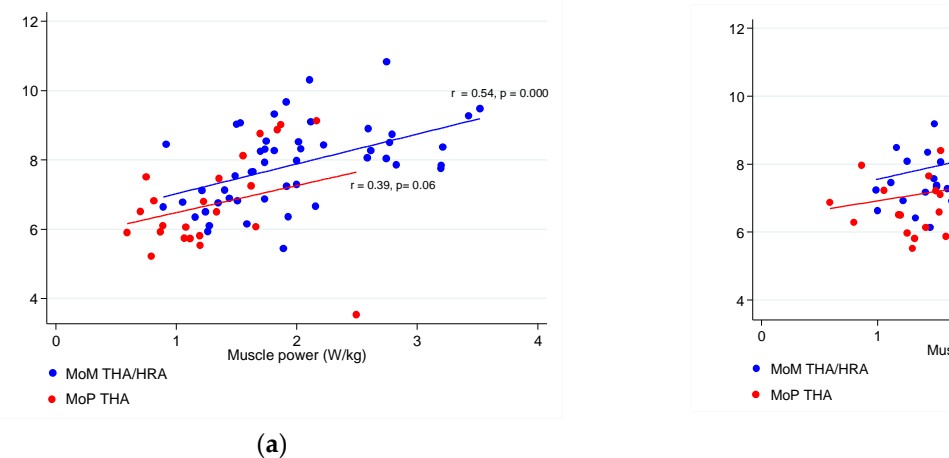
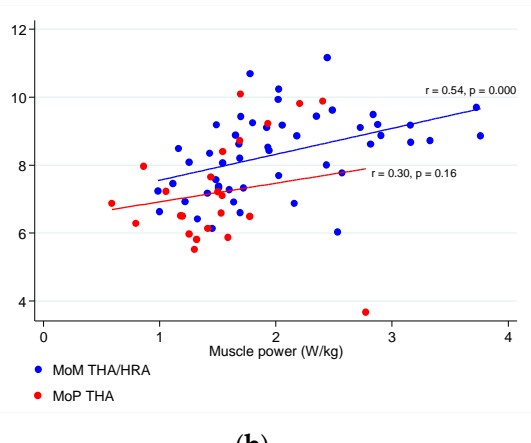

(**a**)　　　　　　　　　　　　　　　　　　　　　　　　　　(**b**)

**Figure 3.** (**a**) Correlations between the mean hip and thigh muscle mass (g/cm$^2$) and the mean muscle power (W/kg) in the implant-side leg of MoM THA/HRA and MoP THA patients. (**b**) Correlations between the mean hip and thigh muscle mass (g/cm$^2$) and the mean muscle power (W/kg) in the non-implant side leg of MoM THA/HRA and MoP THA patients.

### 3.3. Step Test

Step test time asymmetry when ascending was a mean of 11.60 (SD 8.53)% in MoM THA/HRA compared to a mean of 20.60 (SD 16.29)% in MoP THA (*p* = 0.03). When descending, step test time asymmetry was a mean of 13.13 (SD 10.61)% in MoM THA/HRA compared to a mean of 18.81 (SD 14.72)% in MoP THA, but this difference was not significant (*p* = 0.19) (Table 4). Measurements of sagittal and frontal rotation asymmetry when ascending and descending and measurements of sagittal, frontal, and vertical acceleration asymmetry when ascending were also similar between groups (*p* > 0.05) (Table 4).

**Table 4.** Inter-limb difference in trunk Range of Motion in the sagittal and frontal plane measured from step test obtained in the MoM THA/HRA (*n* = 45) and MoP THA (*n* = 19) group, respectively. Values are mean (SD).

| IMU Parameters | MoM THA/HRA | MoP THA | *p*-Value [a] |
|---|---|---|---|
| Step test rotation asymmetry | | | |
| Sagital plane descending (%) | 15.93 (14.86) | 19.24 (11.64) | 0.17 |
| Sagital plane ascending (%) | 19.98 (16.08) | 19.12 (14.74) | 0.95 |
| Frontal plane descending (%) | 17.30 (13.37) | 14.68 (13.34) | 0.39 |
| Frontal plane ascending (%) | 13.78 (11.29) | 15.24 (12.49) | 0.65 |
| Step test time asymmetry | | | |
| Ascending (%) | 11.60 (8.53) | 20.60 (16.29) | 0.03 |
| Descending (%) | 13.13 (10.61) | 18.81 (14.72) | 0.19 |
| Step test acceleration asymmetry | | | |
| Vertical plane ascending (%) | 12.97 (11.38) | 16.64 (15.13) | 0.46 |
| Sagital plane ascending (%) | 17.29 (14.36) | 18.30 (20.39) | 0.56 |
| Frontal plane ascending (%) | 18.07 (13.04) | 21.68 (12.91) | 0.32 |

[a] Two-sample Wilcoxon rank-sum (Mann–Whitney) test.

*3.4. The Harris Hip Score (HHS) and the Copenhagen Hip and Groin Outcome Score (HAGOS) and Correlations*

Scores of the HHS and the HAGOS were similar ($p > 0.05$) for MoM THA/HRA and MoP THA (Table 3). For all patients, significant correlations were found between muscle power (W/kg) and scores of the HHS for the implant-side leg (r = 0.23, $p = 0.04$) and the non-implant-side leg (r = 0.22, $p = 0.05$). This was similar for correlations between muscle power (W/kg) and the HAGOS ADL for the implant-side leg (r = 0.34, $p = 0.003$) and the non-implant-side leg (r = 0.31, $p = 0.008$). Likewise, correlations between muscle power and the HAGOS Sport and recreation for the implant-side leg (r = 0.28, $p = 0.02$) and the non-implant-side leg (r = 0.22, $p = 0.05$) were found. The correlations between muscle power and the other four subscales of HAGOS were not significant ($p > 0.05$) for the implant-side leg and for the non-implant side leg.

*3.5. Sub-Analysis Comparing Males to Males, Females to Females*

Due to the disproportionate MoM/HRA THA and MoP THA groups, we conducted sub analysis comparing females to females (18 MoM/HRA THA vs. 15 MoP THA) and males to males (33 MoM/HRA THA vs. 8 MoP THA) on all reported outcomes (i.e., mean muscle mass, muscle power, patient-reported outcome measures and IMU parameters in the step test). The sub analysis revealed that the inter-limb difference in mean muscle mass in the hip area between MoM THA/HRA and MoP THA was not statistically significant comparing males to males ($p > 0.05$) or females to females ($p > 0.05$) contrary to that reported in Table 2 ($p < 0.05$). However, the other results from the group comparisons did not change with a sub analysis, and thus neither did the overall conclusion.

**4. Discussion**

The purpose of this cross-sectional study was to compare functional outcomes measured as muscle mass, muscle power, and step test asymmetry, between patients undergoing MoM THA/HRA and MoP THA at mean 6.5 years follow-up. Our hypotheses were that MoM THA/HRA would demonstrate superior outcomes in all variables compared to MoP THA, due to the potential advantages of the large femoral heads, and the preoperative selection of younger and presumably more active patients. The first hypothesis was that muscle mass and muscle power was higher in patients undergoing MoM THA/HRA, compared to patients undergoing MoP THA. Our second hypothesis was that the inter-limb difference in muscle mass and muscle power was smaller in patients undergoing MoM THA/HRA, compared to patients undergoing MoP THA. Despite more muscle mass and power in both

legs in patients undergoing MoM THA/HRA, confirming the first hypothesis, the inter-limb differences in muscle mass and muscle power were generally similar between groups. The third hypothesis was that patients undergoing MoM THA/HRA experience less step test asymmetry compared to patients undergoing MoP THA. The step tests results only revealed better performance in ascending as asymmetry was less in the MoM THA/HRA group compared to the MoP THA group, whereas all the other parameters were similar.

The hip muscle mass was similar in the prosthetic hip for both MoM THA/HRA and MoP THA, but the inter-limb difference in hip muscle mass was significantly higher in MoM THA/HRA, meaning that patients undergoing MoM THA/HRA generally had more muscle in the hip region compared with MoP THA. However, patients undergoing MoM THA/HRA in our study had significantly higher muscle power compared to MoP THA in both the implant side leg and the non-implant side leg. This finding was in accordance with DXA findings of more muscle mass in all areas in the MoM THA/HRA group compared to the MoP THA group. These findings have high consistency with previous literature that suggest general lower limb weakness due to reduction in muscle size (i.e., muscle atrophy) [41]. This was in line with our expectations, because the MoM THA/HRA group was generally younger and included more male patients than the MoP THA group, and an age- and sex-related reduction in muscle power has previously been reported [42,43]. The sub analysis revealed that the inter-limb difference in mean muscle mass between MoM THA/HRA and MoP THA was not significant when comparing males to males or females to females. This might be due to type II error as a result of smaller samples. However, the significance of the other reported results had high consistency with the results already reported at overall MoM vs. MoP group levels in the results section.

Other studies have reported MRI-detected muscle atrophy around MoM hip articulations [44–48]. Berber et al. [45] investigated 74 MoM hip arthroplasties, 51 implanted by a posterior approach, 21 by a lateral approach, and 2 unknown. Berber et al. found that 75% had muscle atrophy of gluteus minimus and 50% had muscle atrophy of gluteus medius, with a median MRI-scan interval of 11 months. Likewise, Toms et al. [48] investigated 20 patients with MoM hip arthroplasties, mainly implanted by the posterior approach and a few by the antero-lateral approach, and reported that 45% had muscle atrophy of the gluteus minimus, and 40% had muscle atrophy of the gluteus medius. Besides the theory of muscle atrophy in MoM hip arthroplasties being surgery-induced, some studies suggest that it may be related to an exaggerated inflammatory response to the metal wear debris created by MoM hip articulations, which is thought to reduce function [44,45]. Unfortunately, our data failed to point out which theory is most likely. Rasch et al. [49] also found persistent muscle atrophy around the hip joint two years after MoP THA with a posterior surgical access compared to the contralateral hip.

We chose a step test, which mimics the physical performance of getting into a bus or climbing stairs. Stepping up and down requires balance and a greater range of motion, and is able to identify inter-limb compensations in comparison to level walking [50]. In a step test it is not possible to compensate by shifting weight toward the unaffected leg. Instead, patients compensate with higher trunk flexion and vertical and antero-posterior acceleration to decrease the single leg loading time while stepping up. We found that patients undergoing MoM THA/HRA had significantly less time asymmetry compared to patients undergoing MoP THA when ascending the step bench (11.6% vs. 20.6%). Bolink et al. [31] reported step time asymmetry in 20% of patients with knee osteoarthritis compared to 11% in healthy subjects, and these results from the healthy subjects are similar to our results from MoM THA/HRA. This supports our hypothesis that MoM THA/HRA contributes to superior outcomes in performance-based tests compared to MoP THA. However, it could also reflect that patients undergoing MoM THA/HRA had more muscle power in both legs compared with MoP THA, inherently allowing movements that are more symmetrical and less compensation during stepping. However, our data failed to demonstrate differences in step test rotation and acceleration asymmetry between MoM THA/HRA and MoP THA. This was surprising because the patients undergoing MoM

THA/HRA in our study had more muscle mass and muscle power compared to MoP THA. Furthermore, our MoM THA/HRA group was younger and included more male patients than the MoP THA group. A possible explanation for this could be that the inter-limb difference in muscle mass of the hip and thigh ROI and muscle power were similar between groups, and that the inter-limb difference is of greater importance related to asymmetry than the absolute muscle mass and power found in each leg in performance-based tests. Five patients did not perform the step test. One patient from the MOM HRA group had recently had a knee operation. Four patients experienced balance problems, where one was from the MoM HRA group and three from the MoP THA group. Data from another four patients in the MoM THA/HRA group, and one in the MoP THA group, were missing due to technical problems with the IMU recorder. Thus, the results of the step test remained from 64 patients, 45 in the MoM THA/HRA group and 19 in the MoP THA group. Ten patients did not use the 40 cm step bench, but they accomplished the 30 cm step bench, four in the MoM THA/HRA group and six in the MoP THA group.

The scores for PROMs, HHS, and HAGOS, were similar between the two groups, MoM THA/HRA and MoP THA, which is in line with findings by Varnum et al. [51]. In a nationwide, population-based cross-sectional study, Varnum et al. found similar mean scores when comparing MoM THA/HRA and MoP THA in the PROMs: the five HOOS subscales, the EQ-5D index, EQ VAS score, and the UCLA activity score. Overall, this suggests that MoM THA/HRA is not superior to MoP THA in outcomes that are important to the patients.

The clinical application of the results in this study are that muscle mass and power were reduced in the affected lower limb compared to the non-affected lower limb for both prothesis types, indicating that both groups may benefit from a strength training intervention in their rehabilitation. The inter-limb differences in muscle mass were not statistically significant in thigh-, hip and thigh-, and calf area, but only in the hip area, indicating that patients with MoM/HRA THA may need more specific hip strength training than patients with MoP THA. Furthermore, the study highlights that absolute muscle strength and power may not determine the functional outcome in patients with MoM/HRA THA or MoP THA. The MoM/HRA THA group had more muscle mass in each area compared to the MoP THA group and could produce significantly more muscle power. However, their inter-limb differences or relative muscle strength and power were not significantly different. Therefore, physiotherapists should in the rehabilitation target the resistance training on inter-limb differences in muscle mass or power rather than focusing on absolute muscle mass and power

Our study has several limitations. First, the MoM THA/HRA group was generally younger and included more males than the MoP THA group, and the two groups also differed in years since operation. Age- and gender-matched groups with comparable follow-up time might have changed the results. Even if we could not verify our hypothesis of superior functional outcomes in MoM THA/HRA compared to MoP THA, this does not seem to have affected the conclusion of our results. However, a comparative strength of the groups presented in this study is that patients were of similar nationality, they were included in the same institution with the same indication for surgery, operated by the same surgeons, investigated using the same equipment, and investigated by the same observers. Secondly, the lack of a healthy subject control group prevented comparisons to normative data. Thirdly, our study lacked comparisons with preoperative data, and the fact that any osteoarthritis of the non-prosthetic hip also affects the function. However, we expect that this would bias our results equally in both groups and thus not affect the between-group comparisons. Fourthly, physical and functional capabilities embodying concepts such as muscle strength and power can be confounded by covariates such as sex and age, which are the two covariates we have used to explain differences between the prostheses (MoM/HRA THA vs. MoP THA) [52]. However, inclusion of other covariates such as patient preoperative functional status should have been included to provide a more accurate explanation of the differences [53,54]. Fifthly, DXA-scan is not a gold standard but a reference standard for lean

mass quantification as it cannot quantify fatty infiltration of muscle [55]. Fatty infiltration of muscles after THA is well established in the literature [56–58]. Thus, potential differences or similarities in muscle mass or power between the two prostheses might be due to imprecise quantification of lean mass. However, a strength of the DXA-scan is its high correlation with its more expensive alternatives, MRI-scan and CT-scan, for measures of muscle mass. Sixthly, the effect of the surgical approach (i.e., posterior or antero-lateral) with detachment of different muscles has not been accounted for in this study. Most patients underwent a posterior surgical approach (MoM THA, *n* = 18, MoM HRA THA, *n* = 22 and MoP THA, *n* = 23) and a few underwent an antero-lateral surgical approach (MoM HRA THA, *n* = 11). The evidence of long term implications comparing surgical approaches on clinical, functional, and patient-reported outcomes are sparse [59]. A study found no significant differences in muscle strength between surgical approaches at 3-months follow-up, and another study found no differences in the patient-reported outcome Oxford Hip Score at 5 years follow-up, thus the surgical approach may not have been a significant confounder for muscle strength or patient-reported outcome at mid- to long-term follow-up [60,61]. Finally, our study investigated MoM THA/HRA, the use of which has fallen sharply in clinical practice because of side effects as a result of ARMD. However, the study still contributes with results that can be used to evaluate the effects of a larger femoral head compared to a smaller femoral head.

## 5. Conclusions

Patients undergoing MoM THA/HRA had more muscle mass and leg power compared with MoP THA in the implant side leg and non-implant side leg at mid- to long-term follow-up. In the MoM THA/HRA group, the inter-limb difference in muscle mass in the hip area was higher, which may be related to surgical factors or to an inflammatory response to the metal wear debris. The inter-limb difference in muscle mass was similar in the other regions of interest. Even though the MoM THA/HRA group had more muscle mass and power in both legs, only time asymmetry when ascending during the step test was better in comparison with MoP THA. Overall, we could not verify the hypothesis that MoM THA/HRA contributes to superior functional outcomes in muscle mass, muscle power, physically demanding tests, or patient-reported outcome measures compared to MoP THA. Due to the disproportionate groups, future studies comparing outcome measures between MoM/HRA THA and MoP THA should optimally contain a larger sample size with more homogeneous groups in terms of gender and years since operations, and finally should adjust for the effect of age.

**Author Contributions:** Conceptualization, M.H.H., M.S. and I.M.; methodology, M.H.H., M.S. and I.M.; software, M.H.H.; validation, M.S., M.T., I.M. and F.N.F.; formal analysis, M.H.H.; investigation, M.H.H., M.S., I.M. and M.T.; resources, M.H.H., M.T., I.M., M.T. and F.N.F.; data curation, M.H.H.; writing—original draft preparation, M.H.H.; writing—review and editing, F.N.F.; visualization, M.H.H.; supervision, M.S. and I.M.; project administration, M.H.H.; funding acquisition, M.H.H., M.S. and I.M. All authors have read and agreed to the published version of the manuscript.

**Funding:** This research received funding from Bevica Fonden, Aase and Ejnar Danielsen's Foundation and Zimmer Biomet.

**Institutional Review Board Statement:** The study was conducted in accordance with the Declaration of Helsinki and approved by the Central Denmark Region Committee on Health Research Ethics (17 March 2014; jr. nr.: 1-10-72-65-14) and the Danish Data Protection Agency (17 February 2014; jr. nr.: 2007-58-0010, Trial nr.: 1-16-02-87-14).

**Informed Consent Statement:** Informed consent was obtained from all subjects involved in the study.

**Data Availability Statement:** As part of the Data Use Agreement at the Danish Hip Arthroplasty Registry, authors are not allowed to provide raw data. Upon reasonable request, the corresponding author will provide the statistical programming codes used to generate the results.

**Acknowledgments:** We thank Lone Loevgren, Rikke Moerup, Peter Bo Joergensen, and Inger Krog-Mikkelsen for their help in arranging the follow-up.

**Conflicts of Interest:** The authors declare no conflict of interest.

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
