# Peer review of "A Comparison of Hip Muscle Mass, Muscle Power, and Clinical Outcomes with Long-Term Follow-Up in Patients with Metal-on-Metal Hip Arthroplasty Compared to Metal-on-Polyethylene Hip Arthroplasty"

_applsci, doi:10.3390/app122412772_

Round 1

Reviewer 1 Report

The research is interesting and opens up a field of interest for future interventions and research. It presents an important limitation in the heterogeneity of the sample that is described in the text. 

It is recommended to structure the summary and include a section on conclusions. 

It is recommended to clearly include the objectives of the research in the introduction.

Reviewer 2 Report

In their research, the authors examined the muscle mass, power, gait analysis, and patient reported results of two different total hip arthroplasty techniques (metal-on-metal, MoM, and metal-on-polyethylene, MoP arthroplasty). The purpose was to show that MoM was superior to MoP. The authors were unable to identify any notable difference between the two groups. The two groups, however, were very dissimilar. The two groups really consist of 23 MoP THA and 51 MoM THA/RHA, which vary in terms of sex, age, and years since operation. THA was performed on all MoP patients, but only 35% of MoM patients. When compared to MoP patients, the muscle mass of MoM patients was higher on both the implanted and non-implanted side. Therefore, in my opinion, it is impossible to conclude from this sample that one strategy is preferable than another.

Reviewer 3 Report

I want to congratulate the authors on a quality piece of research.

There are a few minor, cosmetic comments, as figure 1 should be bigger because it loses readability with smaller letters and numbers in table 1 get blended, revise these for a better reading experience.

I understand the premise of this study and the data/results are sound, but explain how such a limitation as a larger, younger, and overly male group is comparable to the MoP group. It’s as if you compare apples to oranges and say even though one group is of orange color they are still round and thus comparable in various attributes. Surely not this much difference, but understand my point.

Males still have more muscle mass and strength compared to females, also patient physique prior to operation needs to be included as a factor as only age does not define „fitness“

-          I see this in figure 1 where readers can see the disproportionate groups include the MoM/MoP sex counts in materials and methods

Also, I would like to see MoM vs MoP comparisons of females to females and males to males.

Statistical analysis needs to take into consideration the points I have made above and try to diminish the disproportion. I leave this to the authors' consideration to address this issue.

Also, try to find studies for discussion if muscle atrophies in MoM or MoP THAs come with age or In older patients or if this is an independent variable.

Round 2

Reviewer 2 Report

In the previous revision, I pointed out that the Authors investigated two very unbalanced groups. They replied with the following sentence: "However, since the demographics favored the MoM THA/HRA group, and we could not verify our hypothesis of superior functional outcomes in MoM THA/HRA patients, this difference does not seem to have favored the conclusion of our results".

I completely disagree with this explanation. What about the circumstance when, with well-balanced groups, the data imply that MoP THA patients get better functional outcomes? This cannot be ruled out.  In my opinion, the conclusion are not supported by the results.

Author Response

Reviewer #2

  1. In the previous revision, I pointed out that the Authors investigated two very unbalanced groups. They replied with the following sentence: "However, since the demographics favored the MoM THA/HRA group, and we could not verify our hypothesis of superior functional outcomes in MoM THA/HRA patients, this difference does not seem to have favored the conclusion of our results". I completely disagree with this explanation. What about the circumstance when, with well-balanced groups, the data imply that MoP THA patients get better functional outcomes? This cannot be ruled out. In my opinion, the conclusion are not supported by the results.

Reply: We agree with the reviewer that balanced groups achieved through matching or randomization is a better design when possible. But, since MoM/HRA hip arthroplasties were generally applied to younger people, a study design with matching or randomized groups was unfortunately not an option. Thus, we did not have this option with the available data/patient number. The comparative strength of the groups (MoM and MoP) presented in this study is that patients were of similar ethnicity and nationality, they were included in the same institution with the same indication for surgery, operated by the same surgeons, investigated using the same equipment, and investigated by the same observers. The impact of unbalanced patient group demographics concerning younger age, more males, shorter follow-up time since surgery in one group (MoM/HRA THA) would expectedly favor the outcomes of functional results in the MoM/HRA THA group. Yet, we found that the MoM/HRA THA group had poorer or equal functional results to the MoP THA group. Therefore, it could be argued, that it makes our conclusion of poorer or equal functional outcomes in the MoM/HRA THA group even stronger.

In addition, in the previous manuscript revision we performed a sub-analysis comparing results of men-to-men and women-to-women. These results only changed the results of the statistical comparison from significant to non-significant for the inter-limb difference in the hip area. However, the results of the sub-analysis for all other outcome measures (mean muscle mass, muscle power, PROMs and IMU parameters) displayed in table 2, 3 and 4 remained unchanged and thus didn’t alter the overall conclusion that MoM/HRA THA and MoP THA had equal results. Thus, we could not conclude that either prosthesis design had superior results. This supports that the disproportion of sex (females/males) between the MoM/HRA THA and the MoP THA groups did not alter the results. We have detailed the specifics of the sub-analysis further in the manuscript. However, we do accept the reviewers skepticism regarding disproportionate groups and the inference of strong conclusions. We have therefore added to the conclusion that future studies needs to address this issue.     

Changes:
Changes Line 208-217
“Sub-analysis comparing males to males, females to females

Due to the disproportionate MoM/HRA THA and MoP THA groups we conducted sub analysis comparing females to females (18 MoM/HRA THA vs 15 MoP THA) and males to males (33 MoM/HRA THA vs 8 MoP THA) on all reported outcomes (i.e., mean muscle mass, muscle power, patient-reported outcome measures and IMU parameters in the Step test). The sub analysis revealed that the inter-limb difference in mean muscle mass in the hip area between MoM THA/HRA and MoP THA was not statistically significant comparing males to males (p > 0.05) or females to females (p > 0.05) contrary to that reported in table 2 (p < 0.05). However, the other results from the group comparisons did not change with a sub analysis, and thus neither did the overall conclusion.”

Changes Line 304-307
“However, a comparative strength of the groups presented in this study is that patients were of similar nationality, they were included in the same institution with the same indication for surgery, operated by the same surgeons, investigated using the same equipment, and investigated by the same observers.”

Changes Line 328-331
“Due to the disproportionate groups, future studies comparing outcome measures between MoM/HRA THA and MoP THA should optimally contain a larger sample size with more homogeneous groups in terms of gender and years since operations and finally adjust for the effect of age.”

Reviewer 3 Report

I think that within possibilities this article was made better, but I still don't like the group disproportion. In this form the article can be published, however, in the future, I would like to see an update with more patients.

Author Response

Reviewer #3

  1. I think that within possibilities this article was made better, but I still don't like the group disproportion. In this form the article can be published, however, in the future, I would like to see an update with more patients.

Reply: Thank you. We agree with the reviewer that balanced groups achieved through matching or randomization is a better design when possible. But, since MoM/HRA hip arthroplasties were generally applied to younger people, a study design with matching or randomized groups was unfortunately not an option. Thus, we did not have this option with the available data/patient number. The comparative strength of the groups (MoM and MoP) presented in this study is that patients were of similar ethnicity and nationality, they were included in the same institution with the same indication for surgery, operated by the same surgeons, investigated using the same equipment, and investigated by the same observers. The impact of unbalanced patient group demographics concerning younger age, more males, shorter follow-up time since surgery in one group (MoM/HRA THA) would expectedly favor the outcomes of functional results in the MoM/HRA THA group. Yet, we found that the MoM/HRA THA group had poorer or equal functional results to the MoP THA group. Therefore, it could be argued, that it makes our conclusion of poorer or equal functional outcomes in the MoM/HRA THA group even stronger.

In addition, in the previous manuscript revision we performed a sub-analysis comparing results of men-to-men and women-to-women. These results only changed the results of the statistical comparison from significant to non-significant for the inter-limb difference in the hip area. However, the results of the sub-analysis for all other outcome measures (mean muscle mass, muscle power, PROMs and IMU parameters) displayed in table 2, 3 and 4 remained unchanged and thus didn’t alter the overall conclusion that MoM/HRA THA and MoP THA had equal results. Thus, we could not conclude that either prosthesis design had superior results. This supports that the disproportion of sex (females/males) between the MoM/HRA THA and the MoP THA groups did not alter the results. We have detailed the specifics of the sub-analysis further in the manuscript. However, we do accept the reviewers skepticism regarding disproportionate groups and the inference of strong conclusions. We have therefore added to the conclusion that future studies needs to address this issue.    

Changes:  
Changes Line 208-217
“Sub-analysis comparing males to males, females to females

Due to the disproportionate MoM/HRA THA and MoP THA groups we conducted sub analysis comparing females to females (18 MoM/HRA THA vs 15 MoP THA) and males to males (33 MoM/HRA THA vs 8 MoP THA) on all reported outcomes (i.e., mean muscle mass, muscle power, patient-reported outcome measures and IMU parameters in the Step test). The sub analysis revealed that the inter-limb difference in mean muscle mass in the hip area between MoM THA/HRA and MoP THA was not statistically significant comparing males to males (p > 0.05) or females to females (p > 0.05) contrary to that reported in table 2 (p < 0.05). However, the other results from the group comparisons did not change with a sub analysis, and thus neither did the overall conclusion.”

Changes Line 304-307
“However, a comparative strength of the groups presented in this study is that patients were of similar nationality, they were included in the same institution with the same indication for surgery, operated by the same surgeons, investigated using the same equipment, and investigated by the same observers.”

Changes Line 328-331
“Due to the disproportionate groups, future studies comparing outcome measures between MoM/HRA THA and MoP THA should optimally contain a larger sample size with more balanced groups in terms of gender and years since operations and finally adjust for the effect of age.”

Round 3

Reviewer 2 Report

Dear Editor,

I would not advise publishing of this work. As previous reviewers have pointed out, the groups are not matched, hence the conclusion cannot be supported by the findings. I have previously rejected this article twice, and the authors' response does not (and cannot) answer my concerns. I thus defer to the Editors' judgment at this time.

Author Response

Response to the manuscript (review round 2) with ID applsci-2009988 “A comparison of hip muscle mass, muscle power and clinical outcomes with long-term follow-up in patients with metal-on-metal hip arthroplasty compared to metal-on-polyethylene hip arthroplasty.”.

We would like to thank the reviewers for their valuable comments and suggestions, which has substantially improved the manuscript. We have tried our best to address the comments and comply with suggested changes the manuscript. The numbered reviewer comments are presented in bold, with our reply below and indicated change/no-change in the manuscript. The line numbering refers to the reversed manuscript. In this response to reviewers changes are marked with italics. In the revised manuscript changes are marked with ‘track changes’.

Comments

  1. Since this special issue is on musculoskeletal rehabilitation, the authors should adequately discuss the practical application of the results of their study within the rehabilitation context.

Reply: We do believe that knowledge of mid- to long-term follow-up on muscle mass, muscle power, functional performance and patient-reported outcomes allows description of characteristics related to each patient group and their prosthetic type (MoM/HRA THA vs MoP THA) as well as a comparison between them. This informs the physiotherapist-led rehabilitation of potential deficits of in patients undergoing MoM/HRA THA and MoP THA at mid- to long term follow-up. It is possible that their musculoskeletal rehabilitation should differentiate by prosthetic type.

Changes: Line 321-333 (Inserted)

“The clinical application of the results in this study are that muscle mass and power was reduced in the affected lower limb compared to the non-affected lower limb for both prothesis types, indicating that both groups may benefit from a strength training intervention in their rehabilitation. The inter-limb differences in muscle mass were not statistically significant in thigh-, hip and thigh- and calf area, but only in the hip area, indicating that patients with MoM/HRA THA may need more specific hip strength training than patients with MoP THA. Furthermore, the study highlights that absolute muscle strength and power may not determine the functional outcome in patients with MoM/HRA THA or MoP THA. The MoM/HRA THA group had more muscle mass in each area compared to the MoP THA group and could produce significantly more muscle power. However, their inter-limb differences or relative muscle strength and power were not significantly different. Therefore, physiotherapists should in the rehabilitation target the resistance training on inter-limb differences in muscle mass or power rather than focusing on absolute muscle mass and power.”   

  1. The limitations of the use of the DXA for lean mass quantification should be clearly stated and discussed in the limitations section. There are in fact a huge literature on the fatty degeneration of muscles in THA and the Authors should at least mention this with respect to the muscle strength and power.
    Furthermore the effects of the surgical approach with detachment of different muscles should also be discussed.

Reply: Thank you for identifying this literature. The limitations of the use of DXA for lean mass quantification have been added and discussed in the limitations section. Furthermore, the presented literature on the fatty degeneration of muscle in THA have been mentioned in the relation to muscle strength and power.    

Changes: Line 355-360 (Inserted)
“Fifthly, DXA-scan is not a gold standard but a reference standard for lean mass quantification as it cannot quantify fatty infiltration of muscle.1 Fatty infiltration of muscles after THA is well established in the litterature.2-4 Thus, potential differences or similarities in muscle mass or power between the two prostheses might be due to imprecise quantification of lean mass. However, a strength of the DXA-scan is its high correlation with its more expensive alternatives MRI-scan and CT-scan for measures of muscle mass.1

Changes Line 360-370 (Inserted)

“Sixthly, the effect of the surgical approach (i.e., posterior or antero-lateral) with detachment of different muscles have not been accounted for in this study. Most patients underwent a posterior surgical approach (MoM THA, n = 18, MoM HRA THA, n = 22 and MoP THA, n = 23) and a few underwent an antero-lateral surgical approach (MoM HRA THA, n = 11). The evidence of long term implications comparing surgical approaches on clinical, functional and patient-reported outcomes are sparse.5 A study found no significant differences in muscle strength between surgical approaches at 3-months follow-up and another study found no differences in the patient-reported outcome Oxford Hip Score at 5 years follow-up, thus surgical approach may not have been a significant confounder for muscle strength or patient-reported outcome at mid- to longterm follow-up.6, 7”      

  1. No information about the prosthesis head dimension is provided. In the introduction it is mentioned  that "MoM THA/HRA with large femoral heads contribute to superior outcome in performance-based tests compared to MoP THA with smaller femoral heads" and in the flow chart different sizes are reported, but no description in the Mat & Met section is provided about this issue, and no relative conclusion has been drawn.

Reply: Thank you for pointing out this inconsistency. However, head sizes are shown in Figure 1. We have now further added a description in the materials and methods section providing femoral head sized in the MoM/HRA THA and MoP THA group. The difference in mean head size is not different between the MoM/HRA THA and MoP THA group: MoM/HRA THA 55.2 [55.3 ; 56.0]

MoP THA            54.6 [53.2 ; 56.2]

Difference          0.53 [-1.1  ; 2.1] p = 0.52

This is due to large variation in the MoP group where 2 large heads increase the mean size for the group. To draw a conclusion on the head size is not possible.

Changes: Line 83-84 (Inserted).
”The MoM/HRA THA group had a mean femoral head size of 55.2 (50-64) and the MoP THA group had a mean femoral head size of 54.6 (50-62). However, femoral head sizes between the two groups are not directly comparable as femoral heads although the same size is larger in MoM/HRA THA compared to MoP THA”

  1. In Figure 2 legend, there is a typing mistake for the (c) and (d) figures. They are both identified as "hip and thigh area". Please correct.

Reply: Thank you for pointing this out. We have made the suggested correction.

Changes: Figure 2 legend:  
“The four different DXA scan regions of interest (ROIs) used for evaluation of mean muscle mass of the implant-side and the non-implant-side leg. (a) calf area, (b) thigh area, (c) hip and thigh area and (d) hip area.”

  1. The information on missing data and heterogeneity of the patients in the performance of the step test reported in lines 134-141 should be mentioned in the limitations section.

Reply: We do agree that these lines represent a limitation. We have moved the suggested section and the preceding sentence (Line 133) to the limitation section.

Changes: Lines 133-141 has been deleted and inserted in Line 306-313 in the reversed manuscript.

”Five patients did not perform the step test. One patient from the MOM HRA group recently had a knee operation. Four patients experienced balance problems, where one was from the MoM HRA group and three from the MoP THA group. Data from another four patients in the MoM THA/HRA group, and one in the MoP THA group were missing due to technical problems with the IMU recorder. Thus, results of the step test remained from 64 patients. 45 in the MoM THA/HRA group and 19 in the MoP THA group. Ten patients did not use the 40 cm step bench, but accomplished the 30 cm step bench. Four in the MoM THA/HRA group and six in the MoP THA group.”   

  1. The order of the sections reported in the methods and in the results sections should be consistent. Please move the "step test" section before the "Harris hip score" section in the results, in accordance with what was done in the methods. In addition, the results related to the Inertial Measurement Unit (IMU) are missing.

Reply: Thank you for pointing out this inconsistency. We have moved the ‘step test’ before the ‘Harris Hip score’ in the results section. We do not agree that the results related to the Inertial Measurement Unit (IMU) are missing. The IMU was used to collect data in the Step test and are thus reported in the Step test section. Step test time-, rotation- and vertical acceleration asymmetry was collected using the IMU in the step test. These results are displayed in Table 4 as well.

Changes: The step test section in the results section has been moved from Line 208-2014 to Line 187-194.

“Step test

Step test time asymmetry when ascending was a mean of 11.60 (SD 8.53) % in MoM THA/HRA compared to a mean of 20.60 (SD 16.29) % in MoP THA (P = 0.03). When descending, step test time asymmetry was a mean of 13.13 (SD 10.61) % in MoM THA/HRA compared to a mean of 18.81 (SD 14.72) % in MoP THA, but this difference was not significant (p = 0.19) (Table 4). Measurements of sagittal and frontal rotation asymmetry when ascending and descending, and measurements of sagittal, frontal, and vertical acceleration asymmetry when ascending were also similar between groups (p > 0.05) (Table 4).”

  1. The sentence reported in lines 253-255 "However...section" is not clear. Please rephrase and give a better explanation on which result has a high consistency with which other result.

Reply: Thank you for pointing out this ambiguity.

Changes: Line 254-262 (Inserted the sentence below. Deleted the previous)

”However, patients undergoing MoM THA/HRA in our study had significantly higher muscle power compared to MoP THA in both the implant side leg and the non-implant side leg. This finding was in accordance with DXA findings of more muscle mass in all areas in the MoM THA/HRA group compared to the MoP THA group. These findings have high consistency with previous literature that suggest general lower limb weakness due to reduction in muscle size (i.e., muscle atrophy).8”   

  1. The sentence reported in lines 304-306 "Since...results" is not clear. Maybe "even if" should be used instead of "since"? Please rephrase.

Reply: Thank you for pointing out this ambiguity. We have made the suggested correction.

Changes: Line 338-340:

Even if we could not verify our hypothesis of superior functional outcomes in MoM THA/HRA compared to MoP THA, this does not seem to have affected the conclusion of our results.”

  1. It is not clear what "constitutes of more factors than sex and age" in line 314 means. Please rephrase the whole sentence (lines 313-315).

Reply: Thank you. We have tried to address this issue by rephrasing the whole sentence as suggested.

Changes: Line 347-352 (Inserted)

”Fourthly, physical- and functional capabilities embodying concepts such as muscle strength and power can be confounded by covariates such as sex and age, which are the two covariates we have used to explain differences between the prostheses (MoM/HRA THA vs MoP THA).9 However, inclusion of other covariates such as patient preoperative functional status should have been included to provide a more accurate explanation of the differences.”

References

  1. Buckinx F, Landi F, Cesari M, et al. Pitfalls in the measurement of muscle mass: a need for a reference standard. https://doi.org/10.1002/jcsm.12268. Journal of Cachexia, Sarcopenia and Muscle. 2018/04/01 2018;9(2):269-278. doi:https://doi.org/10.1002/jcsm.12268
  2. Kovalak E, Özdemir H, Ermutlu C, Obut A. Assessment of hip abductors by MRI after total hip arthroplasty and effect of fatty atrophy on functional outcome. Acta Orthop Traumatol Turc. 2018;52(3):196-200. doi:10.1016/j.aott.2017.10.005
  3. Vadalà AP, Mazza D, Desideri D, et al. Could the tendon degeneration and the fatty infiltration of the gluteus medius affect clinical outcome in total hip arthroplasty? Int Orthop. 2020/02/01 2020;44(2):275-282. doi:10.1007/s00264-019-04468-x
  4. Klemt C, Simeone FJ, Melnic CM, Tirumala V, Xiong L, Kwon Y-M. MARS MRI assessment of fatty degeneration of the gluteal muscles in patients with THA: reliability and accuracy of commonly used classification systems. Skeletal Radiol. 2021/04/01 2021;50(4):665-672. doi:10.1007/s00256-020-03611-9
  5. Petis S, Howard JL, Lanting BL, Vasarhelyi EM. Surgical approach in primary total hip arthroplasty: anatomy, technique and clinical outcomes. Canadian journal of surgery Journal canadien de chirurgie. 2015;58(2):128-139. doi:10.1503/cjs.007214
  6. Winther SB, Husby VS, Foss OA, et al. Muscular strength after total hip arthroplasty. A prospective comparison of 3 surgical approaches. Acta Orthop. 2016;87(1):22-28. doi:10.3109/17453674.2015.1068032
  7. Palan J, Beard DJ, Murray DW, Andrew JG, Nolan J. Which approach for total hip arthroplasty: anterolateral or posterior? Clin Orthop Relat Res. 2009;467(2):473-477. doi:10.1007/s11999-008-0560-5
  8. Loureiro A, Mills PM, Barrett RS. Muscle weakness in hip osteoarthritis: A systematic review. Arthritis care & research (2010). 2013;65(3):340-352. doi:10.1002/acr.21806
  9. Kasper JD, Chan KS, Freedman VA. Measuring Physical Capacity. J Aging Health. 2017;29(2):289-309. doi:10.1177/0898264316635566
